# THE BOLTZMANN POLICY DISTRIBUTION: ACCOUNTING FOR SYSTEMATIC SUBOPTIMALITY IN HUMAN MODELS

**Cassidy Laidlaw**
University of California, Berkeley
cassidy_laidlaw@berkeley.edu

**Anca Dragan**
University of California, Berkeley
anca@berkeley.edu

## ABSTRACT

Models of human behavior for prediction and collaboration tend to fall into two categories: ones that learn from large amounts of data via imitation learning, and ones that assume human behavior to be noisily-optimal for some reward function. The former are very useful, but only when it is possible to gather a lot of human data in the target environment and distribution. The advantage of the latter type, which includes Boltzmann rationality, is the ability to make accurate predictions in new environments without extensive data when humans are actually close to optimal. However, these models fail when humans exhibit *systematic* suboptimality, i.e. when their deviations from optimal behavior are not independent, but instead consistent over time. Our key insight is that systematic suboptimality can be modeled by predicting *policies*, which couple action choices over time, instead of *trajectories*. We introduce the Boltzmann policy distribution (BPD), which serves as a prior over human policies and adapts via Bayesian inference to capture systematic deviations by observing human actions during a single episode. The BPD is difficult to compute and represent because policies lie in a high-dimensional continuous space, but we leverage tools from generative and sequence models to enable efficient sampling and inference. We show that the BPD enables prediction of human behavior and human-AI collaboration equally as well as imitation learning-based human models while using far less data.

## 1 INTRODUCTION

Understanding human preferences, predicting human actions, and collaborating with humans all require models of human behavior. One way of modeling human behavior is to postulate intentions, or rewards, and assume that the human will act rationally with regard to those intentions or rewards. However, people are rarely perfectly rational; thus, some models relax this assumption to include random deviations from optimal decision-making. The most common method for modeling humans in this manner, Boltzmann rationality (Luce, 1959; 1977; Ziebart et al., 2010), predicts that a human will act out a trajectory with probability proportional to the exponentiated return they receive for the trajectory. Alternatively, some human models eschew rewards and rationality for a data-driven approach. These methods use extensive data of humans acting in the target environment to train a (typically high capacity) model with imitation learning to predict future human actions (Ding et al., 2011; Mainprice & Berenson, 2013; Koppula & Saxena, 2013; Alahi et al., 2016; Ho & Ermon, 2016; Ma et al., 2017; Schmerling et al., 2018; Chai et al., 2019; Wang et al., 2019; Carroll et al., 2020).

Both these methods for modeling humans are successful in many settings but also suffer drawbacks. Imitation learning has been applied to modeling humans in driving (Sun et al., 2018), arm motion (Ding et al., 2011), and pedestrian navigation (Ma et al., 2017), among other areas. However, it requires a lengthy process of data collection, data cleaning, and feature engineering. Furthermore, policies learned in one environment may not transfer to others, although there is ongoing work to enable models to predict better out-of-distribution (Torrey et al., 2005; Brys et al., 2015) or in many

---

Our code and pretrained models are available at https://github.com/cassidylaidlaw/boltzmann-policy-distribution.

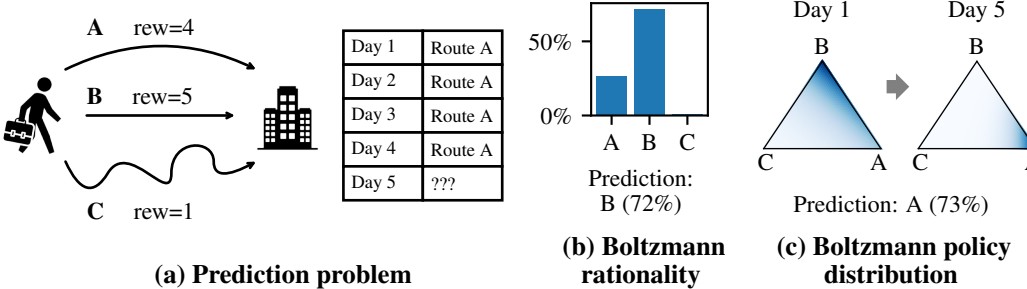

Figure 1: We propose an extension of the Boltzmann rationality human model to a distribution over *policies* instead of trajectories. (a) A person walks to work each day, choosing route A, B, or C and receiving varying rewards based on route length. Route B has the highest reward but for 4 days we observe the person take route A. (b) In this setting, the Boltzmann trajectory model still predicts with high confidence that the person will take route B on day 5 (assuming the reward function is known and cannot be updated). (c) In contrast, our model defines a distribution over policies, the Boltzmann policy distribution (BPD), whose PDF is shown here on the left. Using the BPD as a prior for the person's policy, we can calculate a posterior over policies after the first 4 days (shown on the right) and predict they will take route A with high probability on day 5. In other settings we consider, the BPD adapts in a matter of minutes to a human's policy.

environments simultaneously (Duan et al., 2017; Singh et al., 2020). On the other hand, reward-based models often learn a reward in a low-dimensional space from limited data (Ziebart et al., 2009; Kuderer et al., 2015; Kretzschmar et al., 2016) and tend to transfer more easily to new environments and be robust to distribution shifts (Sun et al., 2021). For many useful tasks, most elements of the reward function might even be clear a priori; in thoses cases, Boltzmann rationality has enabled successful human-AI collaboration based only on data observed during execution of a single task (Bandyopadhyay et al., 2013), e.g. assisting a person to reach an unknown goal (Dragan & Srinivasa, 2013; Javdani et al., 2015). However, Boltzmann rationality fails to capture human behavior that is *systematically* suboptimal. As we show, in some environments this makes it no better at predicting human behavior than a random baseline.

To solve human-AI collaboration tasks, some methods seek to bypass the issues of Boltzmann rationality while still avoiding collecting large amounts of human data. These methods train a cooperative AI policy which is robust to a wide range of potential human behavior via population-based training (Stone et al., 2010; Knott et al., 2021; Strouse et al., 2021), or attempt to find a "natural" cooperative equilibrium, as in the work on zero-shot coordination (Hu et al., 2020; Treutlein et al., 2021). However, such approaches have drawbacks as well. First, they do not define an explicit predictive human model, meaning that they cannot be used to predict human behavior or to infer human preferences. Furthermore, population-based training methods struggle to include enough diversity in the population to transfer to real humans but not so much that it is impossible to cooperate. And, while zero-shot coordination has shown promise in cooperative games such as Hanabi (Hu et al., 2021), it still assumes human rationality and cannot account for suboptimal behavior. Thus, Boltzmann rationality is still widely used despite its shortcomings.

We argue that Boltzmann rationality struggles because it fails to capture systematic dependencies between human actions across time. Consider the setting in Figure 1. Boltzmann rationality predicts the person will take the shortest path, route B, with high probability. Conditioned on the (known) goal of getting to the office, observing the person choose route A gives no information about how the person will act in the future. Thus, the following day, the probability under the Boltzmann trajectory distribution of choosing route A remains unchanged. Similarly, in the setting of Figure 3, observing a person walk to the right around an obstacle gives no information about how they will navigate around it in the future. However, in reality people are consistent over time: they tend to take the same action in the same state, and similar actions in similar states, whether because of consistent biases, muscle memory, bounded computational resources, or myriad other factors. Such consistency is supported well by behavioral psychology (Funder & Colvin, 1991; Sherman et al., 2010).

Our key insight is that systematic suboptimality can be captured by going beyond predicting distributions over *trajectories*, and coupling action choices over time by predicting distributions over *policies* instead. Assuming that people will choose a *policy* with probability proportional to its exponentiated

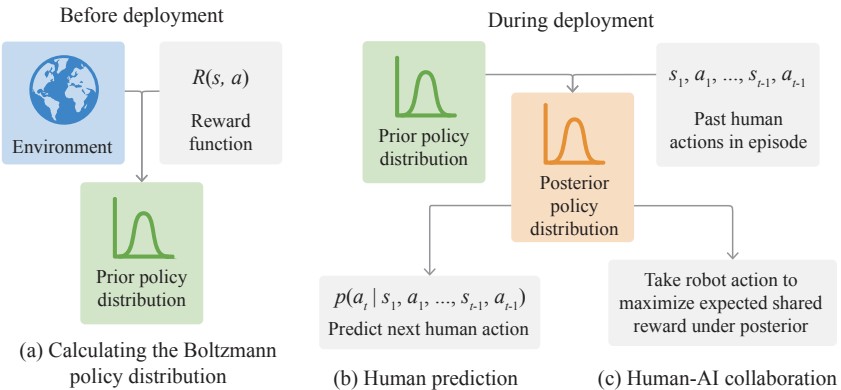

Figure 2: (a) Before deployment, we calculate the Boltzmann policy distribution (BPD) as a prior over human policies by using a model of the environment and the human's reward function. Crucially, this step requires no human data given that we know the human's reward function. Even if the objective is not known, but represented in a low capacity class, it can be inferred via IRL form very limited data in different environments from the target one (Ziebart et al., 2010). Note that using a high capacity model for the reward would become equivalent to imitation learning (Ho & Ermon, 2016) and lose the advantages of Boltzmann rationality. (b) At deployment time, we infer a posterior distribution over a human's policy by updating the BPD prior based on human actions within the episode. This enables adaptive, online prediction of future human behavior. (c) We can also take actions adapted to the posterior distribution in a human-AI collaboration setting.

expected return induces a *Boltzmann policy distribution* (BPD), which we use as a prior over human policies. Observing a person taking actions updates this distribution to a posterior via Bayesian inference, enabling better prediction over time and capturing systematic deviations from optimality. An overview of the process of calculating and updating the BPD is shown in Figure 2. In the example in Figure 1, the BPD predicts that the person will continue to take route A even though route B is shorter. In Figure 3, the BPD predicts that the person is more likely navigate around the obstacle to the right in the future after watching them do it once.

One could argue that updating the BPD in response to observed human actions is no different than training a human model via imitation learning. However, we show that the BPD can give predictive power over the course of a single episode lasting less than three minutes similar to that of a behavior cloning-based model trained on far more data. Thus, the BPD has both the advantages of both Boltzmann rationality—that it works with very little data—as well as imitation learning—that it can adapt to systematically suboptimal human behavior.

One challenge is that calculating the Boltzmann policy distribution for an environment and using it for inference are more difficult than computing the Boltzmann rational trajectory distribution. Whereas the Boltzmann *trajectory* distribution can be described by a single stochastic policy (the maximum-entropy policy), the BPD is a distribution over the space of *all policies*—a high dimensional, continuous space. Furthermore, the complex dependencies between action distributions at different states in the BPD that make it effective at predicting human behavior also make it difficult to calculate.

To approximate the BPD, we parameterize a policy distribution by conditioning on a latent vector—an approach similar to generative models for images such as VAEs (Kingma & Welling, 2014) and GANs (Goodfellow et al., 2014). A policy network takes both the current state and the latent vector and outputs action probabilities. When the latent vector is sampled randomly from a multivariate normal distribution, this induces a distribution over policies that can represent near-arbitrary dependencies across states. See Figure 3 for a visualization of how elements of the latent vector can correspond to natural variations in human policies. In Section 2, we describe how to optimize this parameterized distribution to approximate the BPD by minimizing the KL divergence between the two. During deployment, observed human actions can be used to calculate a posterior over which latent vectors correspond to the human's policy. We investigate approximating this posterior both explicitly using mean-field variational inference and implicitly using sequence modeling.

We explore the benefits of the Boltzmann policy distribution for human prediction and human-AI collaboration in Overcooked, a game where two players cooperate to cook and deliver soup in a

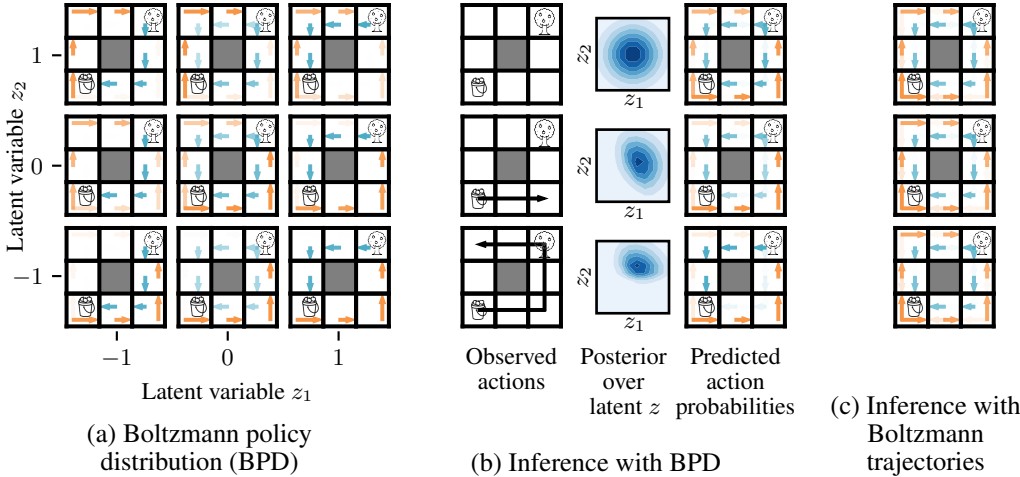

Figure 3: Unlike Boltzmann rationality, the Boltzmann policy distribution captures a range of possible human policies and enables deployment-time adaption based on observed human actions. In this gridworld, the agent must repetitively travel to the apple tree in the upper right corner, pick an apple, and drop it in the basket in the lower left corner. (a) Using the methods in Section 2, we calculate an approximation to the BPD as a policy conditioned on two latent variables $z_1$ and $z_2$. After the distribution is calculated, $z_2$ controls which direction the agent travels on the way to the tree and $z_1$ controls which direction the agent travels on the way back, capturing natural variations in human behavior. (b) After observing 0, 2, and 6 actions in the environment, we show the inferred posterior over $z$ and the resulting predicted action probabilities. The model quickly adapts to predict which way the person will go around the obstacle in the future. (c) In contrast, the Boltzmann trajectory distribution (Boltzmann rationality) cannot adapt its predictions based on observed human behavior.

kitchen (Carroll et al., 2020). The BPD predicts real human behavior far more accurately than Boltzmann rationality and similarly to a behavior cloned policy, which requires vast amounts of human data in the specific target environment. Furthermore, a policy using the BPD as a human model performs better than one using Boltzmann rationality at collaboration with a simulated human; it approaches the performance of a collaborative policy based on a behavior cloned human model.

Overall, the BPD appears to be a more accurate and useful human model than traditional Boltzmann rationality. Further, by exploiting knowledge of the reward function in the task, it achieves performance on par with imitation learning methods that use vastly more data in the target environment. But while in our setting the reward function is known (i.e. the task objective is clear), this is not always be the case. There are many tasks in which it needs to be inferred from prior data (Christiano et al., 2017), or in which reward learning is itself the goal (Ng & Russell, 2000). However, any reward-conditioned human model like the BPD can be inverted to infer rewards from observed behavior using algorithms like inverse reinforcement learning. We are hopeful that using the (more accurate) BPD model will in turn lead to more accurate reward inference given observed behavior compared to using the traditional Boltzmann model.

## 2 THE BOLTZMANN POLICY DISTRIBUTION

We formalize our model of human behavior in the setting of an infinite-horizon Markov decision process (MDP). We assume a human is acting in an environment, taking actions $a \in \mathcal{A}$ in states $s \in \mathcal{S}$. Given a state $s_t$ and action $a_t$ at timestep $t$, the next state is reached via Markovian transition probabilities $p(s_{t+1} \mid s_t, a_t)$. We assume that the person is aiming to optimize some reward function $R : \mathcal{S} \times \mathcal{A} \to \mathbb{R}$. This may be learned from data in another environment or specified a priori; see the introduction for discussion. Rewards are accumulated over time with a discount rate $\gamma \in [0, 1)$.

Let a trajectory consist of a sequence of states and actions $\tau = (s_1, a_1, \ldots, s_T, a_T)$. Let a policy be a mapping $\pi : \mathcal{S} \to \Delta(\mathcal{A})$ from a state to a distribution over actions taken at that state. The Boltzmann rationality model, or Boltzmann trajectory distribution, states that the probability the human will take

a trajectory is proportional to the exponentiated return of the trajectory times a "rationality coefficient" or "inverse temperature" $\beta$:

$$p_{\text{BR}}(\tau) \propto \exp\left\{\beta\sum_{t=1}^{T}\gamma^t R(s_t, a_t)\right\} \tag{1}$$

This is in fact the distribution over trajectories induced by a particular policy, the maximum-entropy policy, which we call $\pi_{\text{MaxEnt}}$ (Ziebart et al., 2010). That is,

$$p_{\text{BR}}(s_1, a_1, \ldots, s_T) = p(s_1) \prod_{t=1}^{T-1} p(s_{t+1} \mid s_t, a_t)\pi_{\text{MaxEnt}}(a_t \mid s_t) \tag{2}$$

Say we are interested in using the Boltzmann trajectory distribution to predict the action at time $t$ given the current state $s_t$ and all previous states and actions. This can be calculated using (2):

$$p_{\text{BR}}(a_t \mid s_1, a_1, \ldots, s_{t-1}, a_{t-1}, s_t) = \frac{p_{\text{BR}}(s_1, a_1, \ldots, s_t, a_t)}{p_{\text{BR}}(s_1, a_1, \ldots, s_t)} = \pi_{\text{MaxEnt}}(a_t \mid s_t) \tag{3}$$

The implication of (3) is that, under the Boltzmann trajectory distribution, the action at a particular timestep is independent of all previous actions given the current state. That is, we cannot use past behavior to better predict future behavior.

However, as we argue, humans are consistent in their behavior, and thus past actions should predict future actions. We propose to assume that the human is not making a choice over *trajectories*, but rather over *policies* in a manner analogous to (1):

$$p_{\text{BPD}}(\pi) \propto \exp\left\{\beta \, \mathbb{E}_{a_t \sim \pi(s_t)}\left[\sum_{t=1}^{\infty}\gamma^t R(s_t, a_t)\right]\right\} = \exp\left\{\beta J(\pi)\right\} \tag{4}$$

(4) defines the Boltzmann policy distribution (BPD), our main contribution. Note that $J(\pi)$ denotes the expected return of the policy $\pi$. Under the Boltzmann policy distribution, previous actions *do* help predict future actions:

$$p_{\text{BPD}}(a_t \mid s_1, a_1, \ldots, s_{t-1}, a_{t-1}, s_t) = \int \pi(a_t \mid s_t) \, p_{\text{BPD}}(\pi \mid s_1, a_1, \ldots, s_{t-1}, a_{t-1}) \, d\pi \tag{5}$$

That is, previous actions $a_1, \ldots, a_{t-1}$ at states $s_1, \ldots, s_{t-1}$ induce a *posterior* over policies $p_{\text{BPD}}(\pi \mid s_1, a_1, \ldots, s_{t-1}, a_{t-1})$. Taking the expectation over this posterior gives the predicted next action probabilities under the BPD.

Both Boltzmann rationality and the BPD assume that the human is near-optimal with high probability, and thus neither may perform well if a person is very suboptimal. However, as we will show in our experiments, people generally *are* close enough to optimal for the BPD to be a useful model.

**Approximating the distribution**    While we have defined the BPD in (4), we still need tractable algorithms to sample from it and to calculate the posterior based on observed human actions. This is difficult because the BPD is a distribution over policies in the high dimensional continuous space $(\Delta(\mathcal{A}))^{|\mathcal{S}|}$. Furthermore, the density of the BPD defined in (4) is a nonlinear function of the policy and induces dependencies between action probabilities at different states that must be captured.

We propose to approximate the BPD using deep generative models, which have been successful at modeling high-dimensional distributions with complex dependencies in the computer vision and graphics literature (Goodfellow et al., 2014; Kingma & Welling, 2014; Dinh et al., 2015). In particular, let $z \sim \mathcal{N}(0, I_n)$ be a random Gaussian vector in $\mathbb{R}^n$. Then we define our approximate policy distribution as

$$\pi(\cdot \mid s) = f_\theta(s, z)$$

where $f_\theta$ is a neural network which takes a state $s$ and latent vector $z$ and outputs action probabilities. Each value of $z$ corresponds to a different policy, and the policy can depend nearly arbitrarily on $z$ via the neural network $f_\theta$. To sample a policy from the distribution, we sample a latent vector $z \sim \mathcal{N}(0, I_n)$ and use it to calculate the action probabilities of the policy at any state $s$. Let $q_\theta(\pi)$ denote this distribution over policies induced by the network $f_\theta$. Figure 3 gives an example of a policy distribution approximated using latent vectors $z \in \mathbb{R}^2$.

To predict the human's next action given previous actions using $q_\theta(\pi)$, it suffices to calculate a posterior over $z$, i.e.

$$q_\theta(a_t \mid s_1, a_1, \ldots, s_{t-1}, a_{t-1}, s_t) = \int f_\theta(s_t, z) \, q_\theta(z \mid s_1, a_1, \ldots, s_{t-1}, a_{t-1}) \, dz \tag{6}$$

We discuss at the end of this section how to approximate this inference problem.

**Optimizing the distribution**    While we have shown how to represent a policy distribution using a

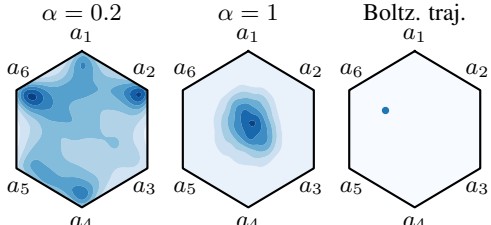 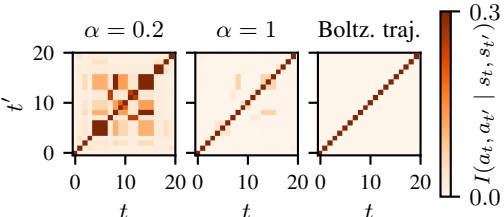

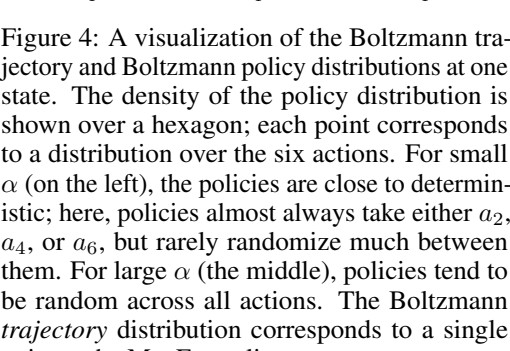

Figure 4: A visualization of the Boltzmann trajectory and Boltzmann policy distributions at one state. The density of the policy distribution is shown over a hexagon; each point corresponds to a distribution over the six actions. For small $\alpha$ (on the left), the policies are close to deterministic; here, policies almost always take either $a_2$, $a_4$, or $a_6$, but rarely randomize much between them. For large $\alpha$ (the middle), policies tend to be random across all actions. The Boltzmann *trajectory* distribution corresponds to a single point at the MaxEnt policy.

Figure 5: A visualization of the mutual information (aka information gain) $I(a_t, a_{t'} \mid s_t, s_{t'})$ between actions taken at different timesteps $t$ and $t'$ given the states at those timesteps under the Boltzmann policy distribution and Boltzmann trajectory distribution (Boltzmann rationality). Unlike Boltzmann rationality, the BPD allows actions taken in the past to help predict actions taken in the future. The information gain is higher the lower the parameter $\alpha$ since more deterministic policies are more predictable.

deep generative model, we still need to ensure that this policy distribution is close to the BPD that we are trying to approximate. To do so, we minimize the KL divergence between the approximation $q_\theta(\pi)$ and $p_{\text{BPD}}(\pi)$:

$$\theta^* = \arg \min_\theta D_{\text{KL}}(q_\theta(\pi) \parallel p_{\text{BPD}}(\pi)) \qquad (7)$$

To optimize (7), we expand the KL divergence:

$$\theta^* = \arg \min_\theta -\mathbb{E}_{\pi \sim q_\theta(\pi)}[\log p_{\text{BPD}}(\pi)] + \mathbb{E}_{\pi \sim q_\theta(\pi)}[\log q_\theta(\pi)]$$

$$= \arg \max_\theta \mathbb{E}_{\pi \sim q_\theta(\pi)}[\beta J(\pi)] + \mathcal{H}(q_\theta(\pi)) \qquad (8)$$

Here, $\mathcal{H}(q_\theta(\pi))$ denotes the entropy of the generative model distribution. Note that although the density $p_{\text{BPD}}(\pi)$ includes a normalization term, it is a constant with respect to $\theta$ and thus can be ignored for the purpose of optimization. Equation (8) has an intuitive interpretation: maximize the expected return of policies sampled from $q_\theta(\pi)$, but also make the policies diverse (i.e., the policy distribution should be high entropy). The first term can be optimized using a reinforcement learning algorithm. However, the second term is intractable to even *calculate* directly, let alone optimize; entropy estimation is known to be very difficult in high dimensional spaces (Han et al., 2019).

To optimize the entropy of the policy distribution, we rewrite it as a KL divergence and then optimize that KL divergence using a discriminator. Let $\mu(\pi)$ denote the base measure with which the density of the BPD is defined in (4). Let $p_{\text{base}}(\pi) = \mu(\pi)/\int d\mu(\pi)$, i.e. $\mu(\pi)$ normalized to a probability measure. Then the entropy of the policy distribution $q_\theta(\pi)$ is equal to its negative KL divergence from $p_{\text{base}}(\pi)$ plus a constant:

$$\mathcal{H}(q_\theta(\pi)) = \log\left(\int d\mu(\pi)\right) - D_{\text{KL}}(q_\theta(\pi) \parallel p_{\text{base}}(\pi)) \qquad (9)$$

We include a full derivation of (9) in Appendix B.1. While (9) replaces the intractable entropy term with a KL divergence, we still need to optimize the divergence. This is difficult as we only have access to samples from $q_\theta(\pi)$, so we cannot compute the divergence in closed form. Instead, we use an adversarially trained *discriminator* to approximate the KL divergence. The discriminator $d$ assigns a score $d(\pi) \in \mathbb{R}$ to any policy $\pi$. It is trained to assign low scores to policies drawn from the base distribution and high scores to policies drawn from $q_\theta(\pi)$:

$$d = \arg \min_d \mathbb{E}_{\pi \sim q_\theta(\pi)}\left[\log\left(1 + \exp\{-d(\pi)\}\right)\right] + \mathbb{E}_{\pi \sim p_{\text{base}}(\pi)}\left[\log\left(1 + \exp\{d(\pi)\}\right)\right] \qquad (10)$$

Huszár (2017) show that if (10) is minimized, then $D_{\text{KL}}(q_\theta(\pi) \parallel p_{\text{base}}(\pi)) = \mathbb{E}_{\pi \sim q_\theta(\pi)}[d(\pi)]$. That is, the expectation of the discriminator scores for policies drawn from the distribution $q_\theta(\pi)$ approximates its KL divergence from the base distribution $p_{\text{base}}(\pi)$. Similar to the training of GANs, the policy network $f_\theta$ attempts to "fool" the discriminator by making it more difficult to tell its

policies apart from those in the base distribution, thus increasing the entropy of $q_\theta(\pi)$.

Putting it all together, the final optimization objective is

$$\theta^* = \arg \max_\theta \mathbb{E}_{\pi \sim q_\theta(\pi)}[\beta J(\pi) - d(\pi)] \tag{11}$$

where $d$ is optimized as in (10). We optimize $\theta$ in (11) by gradient descent. Specifically, at each iteration of optimization, we approximate the gradient of the expectation in (11) with a Monte Carlo sample of several policies $\pi_1, \ldots, \pi_M \sim q_\theta(\pi)$. The gradient of the discriminator score $d(\pi_i)$ can be calculated by backpropagation through the discriminator network. The gradient of the policy return $J(\pi_i)$ is approximated by a policy gradient algorithm, PPO (Schulman et al., 2017).

**Choice of base measure**     An additional parameter to the BPD is the choice of base measure $\mu(\pi)$ and the derived base distribution $p_{\text{base}}(\pi) = \mu(\pi)/\int d\mu(\pi)$. These define the initial weighting of policies before considering the reward function. In our experiments, we define the base measure as a product of independent Dirichlet distributions at each state:

$$\mu(\pi) = p_{\text{base}}(\pi) = \prod_{s \in \mathcal{S}} p_s(\pi(\cdot \mid s)) \qquad p_s \overset{\text{ind.}}{\sim} \text{Dir}(\alpha, \ldots, \alpha) \quad \forall s \in \mathcal{S} \tag{12}$$

The concentration parameter $\alpha$ used in the Dirichlet distribution controls the resulting distribution over policies; we explore the effect of $\alpha$ in Section 2, Figures 4 and 5, and Appendix C.2.

**Parameterization of policy and discriminator**     We modify existing policy network architectures to use as $f_\theta(s, z)$ by adding an attention mechanism over the latent vector $z$. We use a transformer (Vaswani et al., 2017) neural network as the discriminator. See Appendix A.2 for more details.

**Online prediction with the Boltzmann policy distribution**     Predicting actions with the BPD requires solving an inference problem. In particular, we need to solve (6), predicting the next action by marginalizing over the posterior distribution of latent vectors $z$. We explore approximating this inference both explicitly and implicitly. First, we approximate the posterior over $z$ as a product of independent Gaussians using mean-field variational inference (MFVI, see Appendix A) (Blei et al., 2017). This approach, while providing an explicit approximation of the posterior, is slow and cannot represent posteriors which are multimodal or have dependencies between the components of $z$. Our second approach ignores the posterior and instead directly approximates the online prediction problem $q_\theta(a_t \mid s_1, a_1, \ldots, s_t)$. To do this, we generate thousands of trajectories from the BPD by sampling a latent $z$ for each and rolling out the resulting policy in the environment. Then, we train a transformer (Vaswani et al., 2017) on these trajectories to predict the next action $a_t$ given the current state $s_t$ and all previous actions. This avoids the need to compute a posterior over $z$ explicitly.

**Exploring the Boltzmann policy distribution**     Using the methods described above, we compute approximate Boltzmann policy distributions for three layouts in Overcooked: Cramped Room, Coordination Ring, and Forced Coordination. A visualization of the resulting policy distribution for Cramped Room at one state can be seen in Figure 4, which highlights the effect of the parameter $\alpha$. We also visualize the BPD for a simple gridworld environment in Figure 3(a). This figure shows that components of the latent vector $z$ capture natural variation in potential human policies.

In Figure 5, we show that, unlike a Boltzmann distribution over trajectories, the BPD allows actions in previous states to be predictive of actions in future states. The predictive power of past actions is higher for lower values of $\alpha$, which also motivates our choice of $\alpha = 0.2$ for the remaining experiments. Figure 8 shows that the BPD can use the dependence between actions over time to predict systematically suboptimal actions in Overcooked.

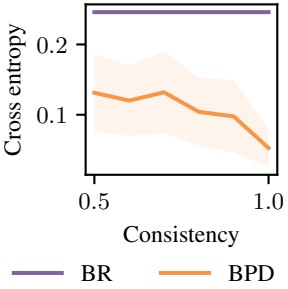

Figure 6: Cross entropy (lower is better) of Boltzmann rationality (BR) and the BPD on synthetic human data in the apple picking gridworld from Figure 3. See Section 3 for details.

## 3   EXPERIMENTS

We evaluate the Boltzmann policy distribution in three settings: predicting simulated human behavior in a simple gridworld, predicting real human behavior in Overcooked, and enabling human-AI collaboration

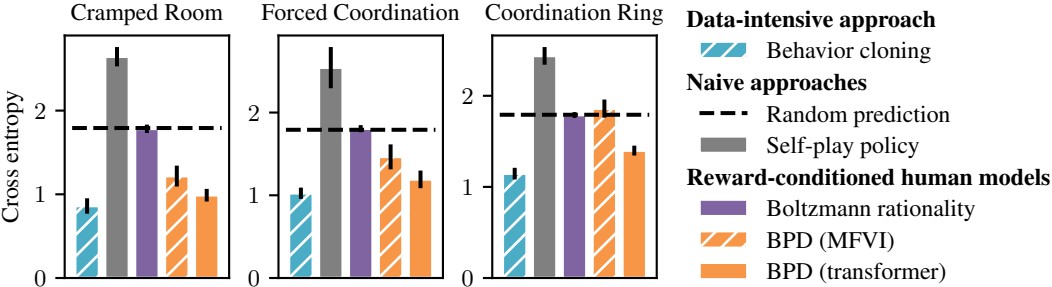

Figure 7: Prediction performance (cross-entropy, lower is better) of various human models on real human data for three Overcooked layouts. Error bars show 95% confidence intervals for the mean across trajectories. See Section 3 for details and discussion.

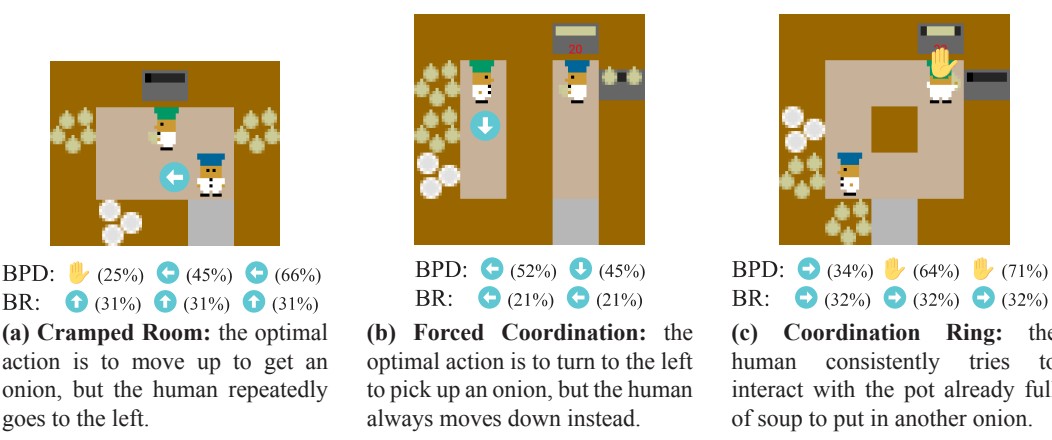

BPD: ✋ (25%) ⬅ (45%) ⬅ (66%)
BR: ⬆ (31%) ⬆ (31%) ⬆ (31%)

**(a) Cramped Room:** the optimal action is to move up to get an onion, but the human repeatedly goes to the left.

BPD: ⬅ (52%) ⬇ (45%)
BR: ⬅ (21%) ⬅ (21%)

**(b) Forced Coordination:** the optimal action is to turn to the left to pick up an onion, but the human always moves down instead.

BPD: ➡ (34%) ✋ (64%) ✋ (71%)
BR: ➡ (32%) ➡ (32%) ➡ (32%)

**(c) Coordination Ring:** the human consistently tries to interact with the pot already full of soup to put in another onion.

Figure 8: The Boltzmann policy distribution predicts real human behavior in Overcooked better than Boltzmann rationality (BR) by adapting to systematically suboptimal humans. In each of the three layouts, the shown state is reached multiple times in a single episode. The human repeatedly takes the suboptimal action overlaid on the game. The predictions of the BPD and BR for each timestep when the state is reached are shown below the game, ignoring "stay" actions (see Appendix C.3). BPD adapts to the human's consistent suboptimality while BR continues to predict an incorrect action. Note that actions at one state can also help predict actions at *different states*; see Figure 5.

in Overcooked. The full details of our experimental setup are in Appendix A. We provide further results and ablations in Appendix C.

**Simulated human data**     First, we investigate if the BPD can predict synthetic human trajectories better than Boltzmann rationality in the apple picking gridworld shown in Figure 3. In this environment, the human must travel back and forth around an obstacle to pick apples and bring them to a basket. To generate simulated human data, we assume the human has a usual direction to travel around the obstacle on the way to the tree and a usual direction to travel on the way back. We assign different simulated humans "consistency" values from 0.5 to 1, and these control how often each human takes their usual direction. A consistency of 0.5 means the human always picks a random direction, while a consistency of 1 means the human always takes their usual direction around the obstacle.

The mean cross-entropy assigned to trajectories sampled from these simulated humans by the BPD and Boltzmann rationality at each consistency value is shown in Figure 6, with the shaded region indicating the standard deviation. As expected, the BPD is particularly good at predicting human behavior for the most consistent humans. However, it even does better than Boltzmann rationality at predicting the humans which randomly choose a direction to proceed around the obstacle each time. This is because our simulated trajectories always go straight to the tree and back to the basket. Boltzmann rationality gives a chance for the human to reverse direction, but the BPD picks up on the consistent behavior and gives it higher probability, leading to lower cross-entropy.

**Human prediction in Overcooked**     Next, we evaluate the predictive power of the Boltzmann

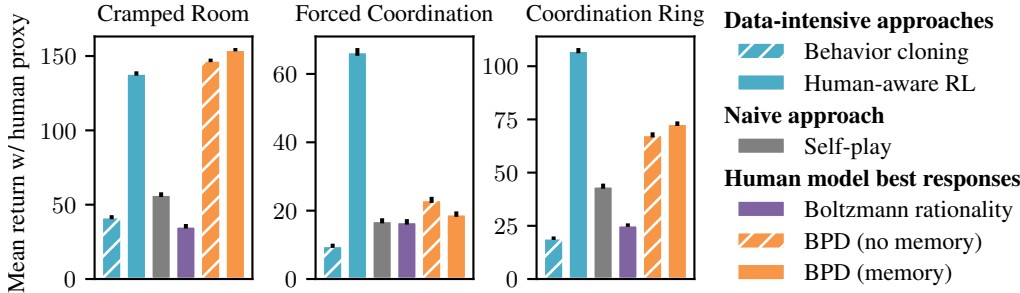

Figure 9: Performance of various "robot" policies at collaborating with a behaviorally cloned "human proxy" policy on three Overcooked layouts. Error bars show 95% confidence intervals for the mean return over trajectories. See Section 3 for details and discussion.

policy distribution on trajectories of real human-with-human play collected by Carroll et al. (2020) in Overcooked. As in the gridworld, we evaluate online prediction: given a human's trajectory until time $t - 1$, predict the action the human will take at time $t$.

We compare three approaches to the BPD for online prediction in Overcooked. First, we train a behavior cloning (BC) model on a set of human trajectories separate from those evaluated on. This model allows us to compare a data-intense prediction method to our approach which adapts over the course of a single trajectory (less than 3 minutes of real time). Second, we train an optimal policy using reinforcement learning (PPO) with self-play. We expect this policy to have low predictive power because humans tend to be suboptimal. Third, we calculate a Boltzmann rational policy.

The predictive performance of all models is shown in Figure 7. The self-play and Boltzmann rational models are both no more accurate than random at predicting human behavior in all three Overcooked layouts. In contrast, the BPD predicts human behavior nearly as well as the behavior cloned model. As expected, the MFVI approach to prediction with the BPD performs worse because its representation of the posterior over $z$ is explicitly restricted. These results validate the BPD as an accurate reward-conditioned human model.

**Human-AI collaboration**    Predicting human behavior is not an end in itself; one of the chief goals of human model research is to enable AI agents to collaborate with humans. Thus, we also evaluate the utility of the Boltzmann policy distribution for human-AI collaboration. The Overcooked environment provides a natural setting to explore collaboration since it is a two-player game. To test human-AI collaborative performance, we pair various collaborative "robot" policies with a simulated "human proxy" policy, which is trained with behavior cloning on held-out human data. Carroll et al. (2020) found that performance this proxy correlated well with real human-AI performance.

We compare several collaborative policies based on similar approaches to the prediction models. First, we evaluate the behavior cloned (BC) model. Since it and the human proxy are both trained from human data, this approximates the performance of a human-human team. Second, we train a human-aware RL policy (Carroll et al., 2020), which is a best-response to the BC policy. This approach requires extensive human data to train the BC policy. We also evaluate the self-play optimal policy; this policy is optimized for collaboration with itself and not with any human model.

Next, we train collaborative policies based on both the Boltzmann rational and BPD models. During each episode of training, the collaborative policy is paired with a simulated human policy from the human model—the MaxEnt policy for Boltzmann rationality, or a randomly sampled policy from the BPD. The human model's actions are treated as part of the dynamics of the environment; this allows the collaborative policy to be trained with PPO as in a single-agent environment. This approach is similar to human-aware RL, but we train with reward-conditioned human models instead of behavior cloned ones. Over the course of training, each collaborative policy maximizes the expected return of the human-AI pair, assuming the human takes actions as predicted by the respective human model.

Because human actions are correlated across time under the BPD, we train BPD-based collaborative policies both with and without memory. The policies with memory have the form $\pi(a_t^R \mid s_1, a_1^H, \ldots, s_{t-1}, a_{t-1}^H, s_t)$; that is, they choose a robot action $a_t^R$ based both on the current state $s_t$ and also human actions $a_{t'}^H$ taken at prior states. This allows adaption to the human's policy.

Figure 9 shows the results of evaluating all the collaborative policies with human proxies on the three Overcooked layouts. In all layouts, the human-aware RL policy does better than either a self-play policy or the BC policy, matching the results from Carroll et al. (2020). The collaborative policies based on Boltzmann rationality tend to perform poorly, which is unsurprising given its weak predictive power. The collaborative policies based on the BPD perform consistently better across all three layouts. Surprisingly, the BPD-based collaborative policies with memory do not perform much differently than those without memory. This could be due more to the difficulty of training such policies than to the inability for adaption using the BPD.

## 4 CONCLUSION AND FUTURE WORK

We have introduced the Boltzmann policy distribution, an alternative reward-conditioned human model to Boltzmann rationality. Our novel model is conceptually elegant, encompasses systematically suboptimal human behavior, and allows adaption to human policies over short episodes. While we focused on human prediction and human-AI collaboration, reward-conditioned human models are also used for reward learning, by inverting the model to infer rewards from behavior. Our results have important implications for reward learning: if Boltzmann rationality is unable to explain human behavior *when the reward is known*, it will be impossible to infer the correct reward given only observed behavior. Since the BPD better predicts human behavior in our experiments, it should lead to better reward learning algorithms as well. We leave further investigation to future work.

ACKNOWLEDGMENTS

We would like to thank Jacob Steinhardt, Kush Bhatia, and Adam Gleave for feedback on drafts, and the ICLR reviewers for helping us improve the clarity of the paper. This research was supported by the ONR Young Investigator Program and the NSF National Robotics Initiative. Cassidy Laidlaw is supported by a National Defense Science and Engineering Graduate (NDSEG) Fellowship.

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

# APPENDIX

## A    EXPERIMENT DETAILS

Here, we give further details about our experimental setup, hyperparameters, and network architectures. We implement the calculation of the BPD and all collaborative training using RLlib (Liang et al., 2018) and PyTorch (Paszke et al., 2019). We use RLlib's PPO implementation with the hyperparameters given in Table 1.

**Gridworld environment**    We implemented the apple-picking gridworld used in Section 3 and Figures 3 and 6. The Boltzmann rational (MaxEnt) policy is calculated with soft value iteration. The Boltzmann policy distribution is calculated using the methods described in Section 2.

**Overcooked environment**    We use the implementation of Overcooked from Carroll et al. (2020). We also use the human data they collected; the train set is used for training the BC policy and the test set is used for training the human proxy policy and evaluating all predictive models. The implementation of Overcooked has changed slightly since the data was collected: players now have to use an "interact" action to start cooking soup. Previously, soup automatically started cooking. We insert"interact" actions in each episode where the players would have had to start cooking the soup to make up for this.

Overcooked is a two-player game while our formalism underlying the BPD applies to single agent environments. However, since Overcooked is perfectly cooperative, the two-player game is equivalent to a single-player game where the agent picks two actions at each timestep independently. Thus, except for when training collaborative policies, we use a single policy network for both players.

We use an episode length of 400 for most experiments; this corresponds to about a minute of real-time play. For human prediction, we use the episode length of the human-human data, which is around 1200 for each episode (about 3 minutes).

Additionally, we implement one small change to the Overcooked environment that we find speeds up reinforcement learning. The environment includes an auxiliary reward with better shaping whose weight is annealed to zero over the course of training. In the original implementation, this auxiliary reward is only given to the agent which takes a particular action, like placing an onion in a pot. We modify this to give the auxiliary reward to both agents such that the environment is fully cooperative. This particularly improves training on Forced Coordination, where one agent is otherwise unable to receive *any* auxiliary reward.

We anneal the auxiliary reward to zero over 2.5 million timesteps, except when calculating the Boltzmann policy distribution. In that case, we anneal over 20 million timesteps because estimating the policy gradient over a distribution of policies results in slower training.

**Calculating the Boltzmann policy distribution**    To calculate the BPD, we use the methods described in Section 2. We set the temperature $1/\beta = 0.1$ for the BPD as well as the Boltzmann trajectory distribution, as we found that this is the maximum that does not lead to near-zero return. We set the concentration parameter of the Dirichlet base distribution described in (12) to $\alpha = 0.2$. We also explore using $\alpha = 1$ in Figures 4 and 5 and give quantitative results in Appendix C.2.

We use latent vectors $z$ of dimension 1,000 for Overcooked. We also tried using a vector of dimension 100, but found this did not give enough flexibility to capture human behavior; see Appendix C.1. We use the same latent vector during each episode for both players.

Since our method for approximating the BPD is similar to GAN training, we find it is prone to instability. Following Radford et al. (2016), we set the Adam momentum term $\beta_1 = 0.5$ when calculating the BPD to reduce this instability.

Overall, we generally found that the BPD required a larger-than-usual RL batch size compared to training a single policy, since the BPD is essentially an infinite population of policies. However, besides this, we were able to use the hyperparameters, network architectures, and reward shaping largely unchanged from the original Overcooked paper. Thus, while the BPD is somewhat more computationally demanding than typical RL methods, we did not find it needed much additional hyperparameter tuning.

**Approximate inference with mean-field variational inference**    Online action prediction with the BPD requires computing a posterior distribution over latent vectors $z$ conditioned on a sequence of past states and actions:

$$q_\theta(z \mid s_1, a_1, \ldots, s_{T-1}, a_{T-1}) \tag{13}$$

We approximate this inference problem with mean-field variational inference (Blei et al., 2017). We approximate the posterior by a parameterized distribution $q_{\mu,\sigma}(z)$ which is a product of independent Gaussians:

$$q_{\mu,\sigma}(z) = \prod_{i=1}^{n} \phi\left(\frac{z_i - \mu_i}{\sigma_i}\right) \tag{14}$$

Here, $\phi(\cdot)$ is the density of a standard normal distribution and $\mu, \sigma \in \mathbb{R}^n$. We minimize the KL divergence between $q_{\mu,\sigma}(z)$ and the true posterior by maximizing the expectation lower bound (ELBO):

$$\max_{\mu,\sigma} \mathbb{E}_{z \sim q_{\mu,\sigma}(z)}\left[\log q_\theta(s_1, a_1, \ldots, s_{T-1}, a_{T-1} \mid z)\right] - D_{\text{KL}}(q_{\mu,\sigma}(z) \parallel \mathcal{N}(0, I_n))$$

$$= \max_{\mu,\sigma} \mathbb{E}_{z \sim q_{\mu,\sigma}(z)}\left[\sum_{t=1}^{T-1} \log f_\theta(a_t \mid s_t, z)\right] - \sum_{i=1}^{n} D_{\text{KL}}(\mathcal{N}(\mu_i, \sigma_i^2) \parallel \mathcal{N}(0, 1)) \tag{15}$$

That is, we optimize $\mu$ and $\sigma$ both (i) to maximize the sum of the log-probabilities assigned to each observed action by policies using latent vectors $z \sim q_{\mu,\sigma}(z)$, and (ii) to minimize KL divergence to the prior over $z$. We use SGD to optimize (15) by estimating the expectation with a Monte Carlo draw of $z$ values and calculating the KL divergence in closed form.

Once the posterior has been approximated by maximizing (15), we can estimate the next action probabilities by marginalizing over the posterior:

$$q_\theta(a_T \mid s_1, a_1, \ldots, s_{T-1}, a_{T-1}, s_T) \approx \mathbb{E}_{z \sim q_{\mu,\sigma}(z)}[f_\theta(a_T \mid s_T, z)]$$

We approximate this expectation via Monte Carlo sampling.

For repeated online prediction over a long episode, performing the optimization in (15) from scratch at every timestep is computationally expensive. We speed up inference by using the $\mu$ and $\sigma$ values from the previous timestep to initialize optimization at the following timestep. Using this initialization, we find we only need one SGD iteration per timestep to effectively optimize (15).

**Training the predictive sequence models**    As describe in Section 3, we also approximate the online action prediction problem

$$q_\theta(a_t \mid s_1, a_1, \ldots, s_{t-1}, a_{t-1}, s_t) \tag{16}$$

with the BPD by training a transformer on rollouts from the distribution. Specifically, we sample 50,000 policies from the BPD and roll out a trajectory for each policy. Then, we train both a transformer and an LSTM (Hochreiter & Schmidhuber, 1997) to predict (16) over this set of trajectories by standard supervised learning. We describe the specific training hyperparameters in Table 2 and architectures in Appendix A.2.

**Behavior cloning**    We use a similar behavior cloning (BC) procedure to Carroll et al. (2020). BC policies are trained to minimize the cross entropy over human trajectories with Adam (Kingma & Ba, 2014). We lower the learning rate by a factor ten if the cross entropy has not meaningfully improved in 5 epochs. We regularly evaluate the BC policies playing with themselves in Overcooked. We chose an iteration to stop training for each layout based on when the policies perform best. We use the manually designed features from Carroll et al. (2020) as input to the BC policy network. They also hardcode their BC policies to take a random action when stuck in the same state for too long; we do not do this.

**Training the collaborative policies**    As described in Section 3, we train collaborative policies as a best response to the BC policy (i.e., human-aware RL), the Boltzmann rational (MaxEnt) policy, and the BPD. We train the collaborative policies using PPO for one player while using a fixed policy for the other player.

Since the BPD enables better prediction of future behavior given past behavior, we tried training a policy with memory to cooperate with policies sampled from BPD. This policy takes as input not only the current state, but also previous states and the human actions at those states:

$$\pi(a_t^R \mid s_1, a_1^H, \ldots, s_{t-1}, a_{t-1}^H, s_t)$$

We found that directly trying to optimize such a policy with PPO did not lead to good performance. Instead, we use the hidden state of the LSTM network trained for online prediction as an extra input to the policy network (in addition to the state). We hypothesize that the LSTM's hidden state already contains all necessary information to adapt to the human's policy, so we do not need to further train the LSTM during policy optimization. We find that this leads to much more stable training. However, learning a policy with memory is still slower, so we use more training iterations of PPO (see Table 1).

**Evaluation collaborative performance** We evaluate the performance of collaborative policies with a human proxy policy behavior-cloned from held-out data. Since each Overcooked layout has different starting positions for the two players, we roll out 1,000 episodes with each starting position permutation and average all the returns.

## A.1 HYPERPARAMETERS

| PPO Hyperparameter | Value (Gridworld) | Value (Overcooked) |
|---|---|---|
| Training iterations | 500 | 500 |
| (policies with memory) | | 1,000 |
| Batch size | 2,000 | 100,000 |
| SGD minibatch size | 2,000 | 8,000 |
| SGD epochs per iteration | 8 | 8 |
| Optimizer | Adam | Adam |
| Learning rate | $10^{-3}$ | $10^{-3}$ |
| Gradient clipping | 0.1 | 0.1 |
| Discount rate ($\gamma$) | 0.9 | 0.99 |
| GAE coefficient ($\lambda$) | 0.98 | 0.98 |
| Entropy coefficient | 0 | 0 |
| KL target | 0.01 | 0.01 |
| Clipping parameter ($\epsilon$) | 0.05 | 0.05 |

Table 1: PPO hyperparameters.

| Hyperparameter | Value (Overcooked) |
|---|---|
| Training epochs | 4 |
| Batch size (num. episodes) | 40 |
| Optimizer | Adam |
| Learning rate | $10^{-3}$ |

Table 2: Sequence model training hyperparameters.

| BC Hyperparameter | Value (Overcooked) |
|---|---|
| Training epochs | |
| (Cramped Room) | 500 |
| (Forced Coordination) | 150 |
| (Coordination Ring) | 250 |
| SGD minibatch size | 64 |
| Optimizer | Adam |
| Initial learning rate | $10^{-3}$ |

Table 3: Behavior cloning hyperparameters.

## A.2 NETWORK ARCHITECTURES

Here, we describe the architectures used for policy networks, the discriminator from Section 2, and sequence models from Section 3.

**Behavior cloning**  For our behavior cloned policies in Overcooked, we use a multilayer perceptron (MLP) with two hidden layers of size 64.

**Overcooked policies**  For all other policies in Overcooked, we input the state as a two-dimensional grid with 26 channels corresponding to various features in the environment, e.g. counters, players, etc. These states are first processed through a convolutional neural network with three layers of 25 hidden units each and filter sizes of (5, 5), (3, 3), and (3, 3) respectively. Then, the resulting activations are flattened and passed through three fully-connected layers with 64 hidden units. We use leaky ReLUs with negative slope -0.01 for all activation functions.

**Conditional policy network**  The conditional policy network $f_\theta(s, z)$ used to approximate the BPD additionally takes in as input $z \in \mathbb{R}^n$, the random latent vector. We found that directly concatenating $z$ to the state representation or the input of the first fully-connected layer led to slow and unstable training. Instead, we used an attention mechanism (Bahdanau et al., 2016) over the latent vector. Let $a$ be the activations after the first fully-connected layer. We transform $a$ by a fully connected layer to a matrix of size $m \times n$ and then take the softmax over each row such that the elements add to 1. This gives a matrix $W$, our attention weights. Then, we concatenate $Wz$ to $a$ and feed the resulting vector through the remainder of the network. This allows the network to attend to arbitrary combinations of the values in $z$ and improves training considerably. We set $m = 4$ for the gridworld experiments and $m = 10$ for Overcooked.

**Discriminator**  We give as input to the discriminator network $d$ a sequence of states and actions $s_1, a_1, \ldots, s_k, a_k$. We let $k = 10$, finding that using more actions and states leads to instability because it is too easy for the discriminator to tell apart policies from the base distribution and the approximated BPD.

The actions at each state are one-hot encoded and appended to the state representation. Then, for Overcooked, the states are passed through a convolutional state encoder, which creates a separate representation for each state-action pair. Next, the resulting representations are passed through a three-layer transformer with one head and $d_{\text{model}} = 64$. This allows the discriminator to pool information about the policy across states. Finally, the outputs of the transformer are averaged and fed through a final linear layer to produce the discriminator score of the policy.

**Sequence models for prediction**  As described above, we train both LSTMs and transformers to estimate (16), i.e. to predict the next human action $a_T$ given the current state $s_t$ and past states and actions $s_1, a_1, \ldots, s_{T-1}, a_{T-1}$. As input to these sequence models, we give a tuple of $(s_{t-1}, a_{t-1}, s_t)$ at each timestep $t$, and train them by minimizing the cross entropy between the output and the true action $a_t$. We find that giving as input both the previous state $s_{t_1}$ and previous action $a_{t-1}$, as opposed to just the previous action $a_{t-1}$, helps the models to better associate past actions with the states in which they were taken.

The remainder of a the sequence model for Overcooked consists of a policy network as described above, with the fully connected layers replaced by either an LSTM or transformer. We use 3 layers and 256 hidden neurons for both the LSTM and transformer.

# B  DERIVATIONS

## B.1  ENTROPY AS KL DIVERGENCE

Here, we derive the equation used in Section 2 to relate the entropy of the policy distribution $q_\theta(\pi)$ to its KL divergence from the normalized base measure $p_{\text{base}}(\pi) = \mu(\pi) / \int d\mu(\pi)$. First, note that

$$\frac{d\mu}{dp_{\text{base}}}(\pi) = \int d\mu(\pi),$$

where $\frac{d\mu}{dp_{\text{base}}}$ is the Radon-Nikodym derivative between $\mu(\pi)$ and $p_{\text{base}}(\pi)$. Now,

$$
\begin{aligned}
\mathcal{H}(q_\theta(\pi)) &= \int - \log \frac{dq_\theta}{d\mu}(\pi) \; dq_\theta(\pi) \\
&= \log\left(\int d\mu(\pi)\right) - \int \log\left(\int d\mu(\pi)\right) + \log \frac{dq_\theta}{d\mu}(\pi) \; dq_\theta(\pi) \\
&= \log\left(\int d\mu(\pi)\right) - \int \log\left(\frac{dq_\theta}{d\mu}(\pi) \int d\mu(\pi)\right) \; dq_\theta(\pi) \\
&= \log\left(\int d\mu(\pi)\right) - \int \log\left(\frac{dq_\theta}{d\mu}(\pi) \frac{d\mu}{dp_{\text{base}}}(\pi)\right) \; dq_\theta(\pi) \\
&= \log\left(\int d\mu(\pi)\right) - \int \log\left(\frac{dq_\theta}{dp_{\text{base}}}(\pi)\right) \; dq_\theta(\pi) \\
&= \log\left(\int d\mu(\pi)\right) - D_{\text{KL}}(q_\theta(\pi) \,\|\, p_{\text{base}}(\pi)).
\end{aligned}
$$

## B.2 DISCRIMINATOR APPROXIMATES KL DIVERGENCE

For completeness, we also derive the result of Huszár (2017) showing that minimizing the discriminator objective given in (10) approximates the KL divergence between $q_\theta(\pi)$ and $p_{\text{base}}(\pi)$. Recall that the discriminator is chosen by solving the following optimization problem:

$$
d = \arg\min_d \; \mathbb{E}_{\pi \sim q_\theta(\pi)}\Big[ \log\left(1 + \exp\{-d(\pi)\}\right) \Big] + \mathbb{E}_{\pi \sim p_{\text{base}}(\pi)}\Big[ \log\left(1 + \exp\{d(\pi)\}\right) \Big]. \quad (17)
$$

Assuming that $q_\theta(\pi)$ and $p_{\text{base}}(\pi)$ both have a density with respect to the Lebesgue measure, we can rewrite (17) as

$$
d = \arg\min_d \; \int \log\left(1 + \exp\{-d(\pi)\}\right) q_\theta(\pi) + \log\left(1 + \exp\{d(\pi)\}\right) p_{\text{base}}(\pi) \; d\pi. \quad (18)
$$

Clearly, the integral in (18) is minimized if and only if the integrand is minimized for all $\pi$. That is, we have that

$$
\forall \pi \quad d(\pi) = \min_{d(\pi) \in \mathbb{R}} \log\left(1 + \exp\{-d(\pi)\}\right) q_\theta(\pi) + \log\left(1 + \exp\{d(\pi)\}\right) p_{\text{base}}(\pi).
$$

It is straightforward to show that the value of $d(\pi)$ which minimizes this is

$$
d(\pi) = \log\left(\frac{q_\theta(\pi)}{p_{\text{base}}(\pi)}\right).
$$

Thus, taking the expectation of $d(\pi)$ gives the KL divergence between $q_\theta(\pi)$ and $p_{\text{base}}(\pi)$:

$$
\mathbb{E}_{\pi \sim q_\theta(\pi)}[d(\pi)] = \mathbb{E}_{\pi \sim q_\theta(\pi)}\left[\log\left(\frac{q_\theta(\pi)}{p_{\text{base}}(\pi)}\right)\right] = D_{\text{KL}}(q_\theta(\pi) \,\|\, p_{\text{base}}(\pi)).
$$

## C ADDITIONAL EXPERIMENTS

In this appendix, we present results from additional experiments that did not fit in the main text. First, we explore alternative choices of hyperparameters for the BPD. We also include an ablation of prediction with sequence models using the BPD. We conclude with a closer look at prediction of human behavior in Overcooked and what makes different human models perform better or worse.

### C.1 LATENT VECTOR DIMENSION

We explore the effect of altering the dimensionality $n$ of the latent vector $z$ used to approximate the BPD through the policy network $f_\theta(s, z)$. We use $n = 1,000$ throughout the rest of our experiments, but we also tried using $n = 100$. The effects on prediction and collaboration are shown in Figure 10. We find that reducing $n$ has negative effects on collaborative performance and also on prediction using MFVI. Surprisingly, setting $n = 100$ does not affect the performance of a transformer trained to perform prediction using the BPD. We are not sure why this happens—the transformer may simply be good at predicting out-of-distribution. Even if explicitly optimize over values of $z$ for low cross entropy on human data when $n = 100$, we are unable to reach the performance of the transformer.

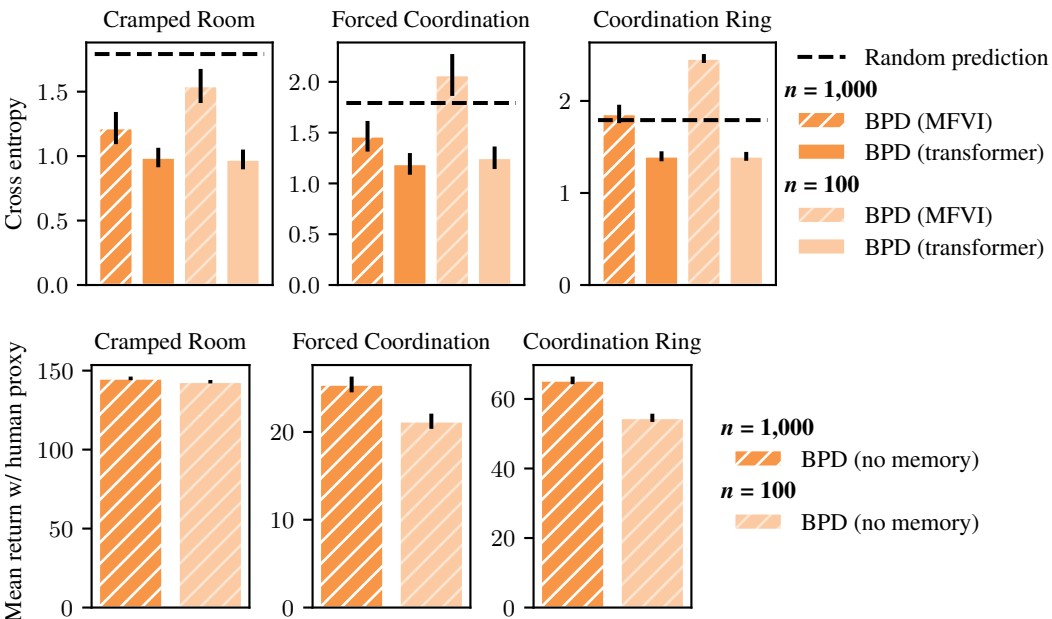

Figure 10: The effect on human prediction and human-AI collaboration of changing the dimension $n$ of the latent vector $z$ used to approximate the BPD. See Appendix C.1 for details and discussion.

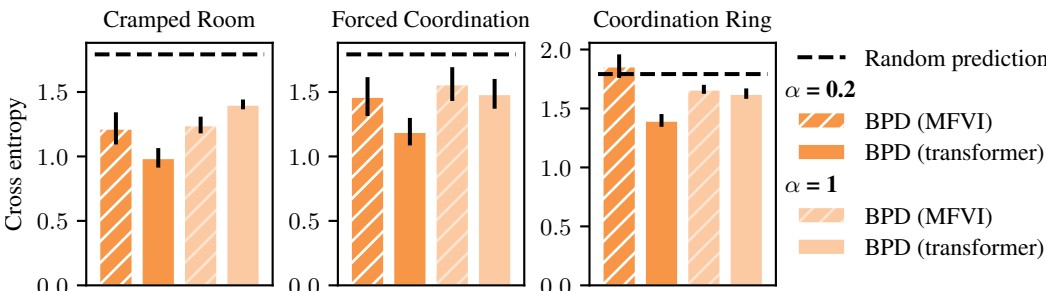

Figure 11: The effect on human prediction of changing the concentration $\alpha$ of the Dirichlet base distribution $p_{\text{base}}(\pi)$ using to calculate the BPD. See Appendix C.2 for details and discussion.

## C.2 BASE DISTRIBUTION CONCENTRATION

Next, we try changing $\alpha$, the parameter of the Dirichlet distribution used to define $p_{\text{base}}(\pi)$. Recall from Section 2 and Figures 4 and 5 that lower values of $\alpha$ correspond to a more consistent human while higher values correspond to a more random human. We use $\alpha = 0.2$ for most experiments throughout the paper, but here we explore the effects of using $\alpha = 1$ for predicting human behavior. The results are shown in Figure 11. Using a transformer with $\alpha = 0.2$ predicts human behavior the best for all layouts, validating our choice of $\alpha$ and our assumption that humans are consistent.

## C.3 FURTHER PREDICTION EXPERIMENTS

Here, we investigate what makes different models better or worse at human prediction in Overcooked. In particular, we noticed that the majority of the actions taken in Overcooked by humans are "stay" actions—that is, actions that do not do anything. Thus, simply predicting a "stay" action with high probability can lead to good predictive performance. In Figure 12 we separate out the stay actions and, in the right two columns, show the cross entropy and accuracy of models excluding these actions. That is, we remove all timesteps when a stay action was taken and remove the stay probability from the output of the human model. After removing stay actions, Boltzmann rationality is a better predictor of human behavior, although behavior cloning and the BPD are still superior. We also note

that while the BPD has similar *cross entropy* to behavior cloning, its *accuracy* is slightly lower on non-stay actions. This could explain why collaborative policies trained with the BPD do not always outperform those trained with a behavior cloned human model, e.g. on Forced Coordination (see Figure 9).

We also explored some other baselines and ablations of the BPD. First, we try removing all reward information and simply trying to predict based on a random distribution over policies—that is, using a uniform prior over policies and doing online inference. Using the same procedure as for the BPD, we sample trajectories from thousands of random policies drawn from $p_{\text{base}}(\pi)$ and train a transformer for online prediction over these trajectories. The resulting prediction performance on real human data is shown in Figure 12 as the white bars. At first, it appears that this technique is quite successful, rivaling behavior cloning and the BPD in cross entropy and accuracy. However, this largely seems to be because it consistently predicts the human will take a stay action. Removing these actions, it is clear that using a random policy distribution does not results in good predictive performance. These experiments validate the utility of knowing the human's reward function for predicting human behavior and serve as an ablation of our method of online prediction.

Next, we explored alternatives to behavior cloning. As discussed in Appendix A, we used hand-crafted features to train all the behavior cloning models, following Carroll et al. (2020). We additional tried BC directly using the raw observations and found it performed slightly worse. Furthermore, we explored using generative adversarial imitation learning (GAIL) (Ho & Ermon, 2016) as a more robust alternative to BC. We found that both these approaches performed similarly to or slightly worse than BC; see Figure 12 for the full results.

Finally, we explored an alternative to online prediction with the BPD. For this baseline, we simply started with a random initialized policy network and updated it after each timestep of a test episode using the BC loss. The results for this method using either hand-crafted features or raw observations are shown in Figure 12 with the label "Online BC." Online BC is directly comparable to online inference with the BPD since they both do not require prior human data. However, we generally found that online BC does not perform as well as BC or the BPD, particularly when using raw observations.

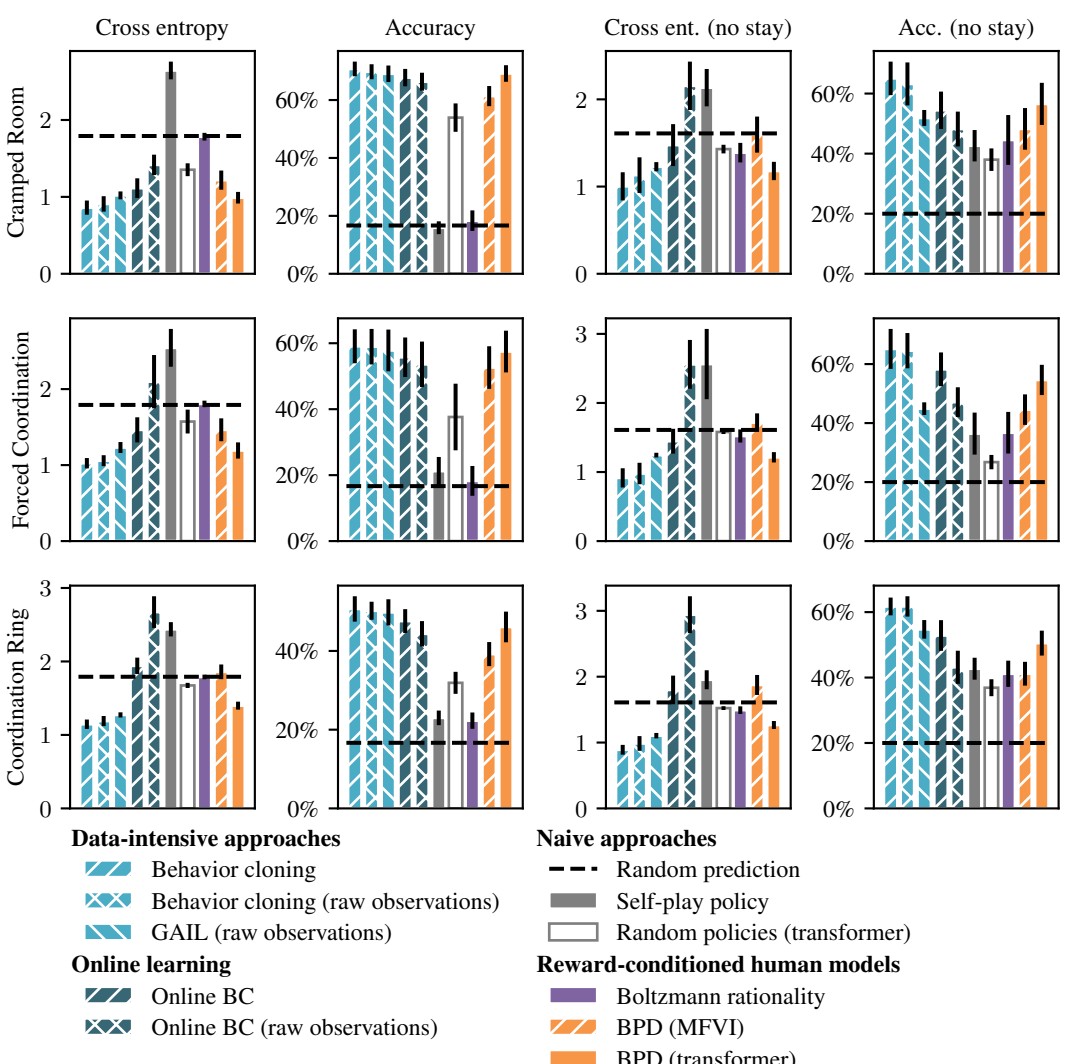

Figure 12: A variety of metrics for predictive power of various human models on the three Overcooked layouts. Lower cross entropy and higher accuracy is better. The right two columns exclude "stay" actions, which make up the majority of the human data, from consideration. See Appendix C.3 for details and discussion.

