# OpenReview forum: "The Boltzmann Policy Distribution: Accounting for Systematic Suboptimality in Human Models"
_ICLR.cc/2022/Conference — ICLR 2022 Poster_

### Official Review · Reviewer_Bv5s · 2021-10-31

**Correctness:** 4
**Technical Novelty And Significance:** 3
**Empirical Novelty And Significance:** 3
**Recommendation:** 6
**Confidence:** 3

**Main Review:**


* Pros
    * Nice writing and figures.
    * Very clear difference between Boltzmann rational model and the proposed method.
    * Real world (Overcooked game) dataset evaluation.
    * The authors not only evaluated prediction power of the proposed method in Overcooked, but also combine the proposed method with RL agent to do the Overcooked game.

* Majors
    * I think (please correct me if I am wrong :) the proposed approach is very much similar to GAIL (`Ho, J., & Ermon, S. (2016). Generative adversarial imitation learning. Advances in neural information processing systems, 29, 4565-4573.`)
        * The generator, `f_\theta` in this work = `\pi_\theta` in Alg.1 in GAIL.
        * The discriminator, `d` in this work = `D_w` in Alg.1 in GAIL.
        * The sampled path, `q_\theta = {s1,a1,…,s_{t-1},a_{t-1}}` in this work = `\tau_i ~ \pi_\theta_i`, in Alg.1 in GAIL.
        * The base measure, `p_base` in this work = the expert policy, `\tau_E ~ \pi_E`, in Alg.1 in GAIL.
        * In my opinion, this work is similar to GAIL. The only differences are the followings:
            * This work captures the suboptimality in the human behavior in the base measure, while GAIL assumes that the human is optimal.
              * This work's reward is implicitly hidden in the base measure, while GAIL's reward can be interpreted as `D_w`.
        * If this is true, it would be great if the authors could add some text to explain the relation between the proposed method and GAIL. This could be formally and/or empirically comparison.


* Minor:
    * The comparison between the proposed method and behavior cloning is a bit unfair, since the proposed method has the knowledge about the human reward, while the behavior cloning does not.
    * Page 2: `Conditioned on the (known) goal of getting to the office, observing the person choose route A gives no information about how the person will act in the future.`
      * This sentence makes sense assuming that the human reward function is correct and fixed. I think if the human reward can be updated, then observing that the human choosing route A could provide information. It would be great to clarify this here.
    * Page 5: `We discuss in Section 3 how to approximate this inference problem.`
      * Section 3 is a typo?
    * Page 6: Eq9: `p_unif` should be `p_base`?


**Summary Of The Paper:**


* This paper proposes an interesting approach to model and predict human behavior.

* The approach is called Boltzmann policy distribution (BPD). It improves the Boltzmann rational model in that BPD considers the systematic suboptimality in human behavior.
    * Systematic suboptimality means that the human could be consistent in producing suboptimal behavior. Hence, to capture systematic suboptimality, it is necessary to combine the human's reward function (optimal behavior) and trajectory data (deviation from optimality) in human modeling and prediction.

* This approach predict human policies, rather than trajectories, so that it can capture the (systematically suboptimal) human behavior that is reflected in the human action choices over time.

* Approach detail: the approach follows GAN (generative adversarial networks):
    * The goal is to compute the human BPD policy `= \integral \pi(a|s) * p_PBD(\pi | s1,a1,...,s_{t-1},a_{t-1})` (Eq5).
        * `\pi(a|s)` is approximated as a **generator**, `f_\theta(s,z)` (Eq.6).
        * `p_PBD(\pi | s1,a1,...,s_{t-1},a_{t-1})` is approximated as sampled human trajectories, `q_\theta(z | s1,a1,...,s_{t-1},a_{t-1})` (Eq.6).
    * The base measure, `p_base(\pi)`,  is defined as the optimal human behavior based on the known human reward function with suboptimality parameter \beta (Eq4).
    * The **discriminator**, `d`, used to distinguish between the policy generated by the base measure, `p_base(\pi)`, and the policy generated by the sampled human trajectories, `q_\theta(\pi)`.
    * The networks can be seen as: `\pi(s) -> d(\pi) -> z -> f_\theta(s,z) -> \pi(s) reconstructed`.
    * The training for the generator, `f_\theta(s,z)`, is in Eq8,9 and the training for the discriminator, `d`, is in Eq10.


* Results
    * Apple picking gridworld with simulated human data
        * Cross entropy: BPD < Boltzmann rational models
    * Overcooked (prediction only) with real human-with-human play data
        * Cross entropy: BPD = behavior cloning < random < PPO+self-play, Boltzmann rational models
    * Human-AI collaboration (prediction and control in Overcooked) with "human proxy" policy from previous work
        * Mean return: human-aware RL policy (prev work) > policy based on Boltzmann rationality, self-play policy, behavior cloning policy
        * Mean return: policy based on BPD > policy based on Boltzmann rationality, self-play policy, behavior cloning policy
        * Mean return: policy based on BPD > Human-aware RL policy (prev work) in 2 out of 3 cases.


**Summary Of The Review:**

* I think it would be great if the connection between the proposed method and GAIL could be stated more clearly (if there are any). And empirical comparison with GAIL could be helpful, too. This is why I currently give 2 for the `2: The contributions are only marginally significant or novel.`

# Post-rebuttal
Thank you for your clarification! I have adjusted my score.

---

> ### Author Response · Authors · 2021-11-16
> **Response to Reviewer Bv5s**
>
> We thank the reviewer for their extensive comments. We are glad the reviewer found our paper well written with a clear difference demonstrated between Boltzmann rationality and our proposed BPD model. Below, we have addressed the concerns raised by the reviewer.
>
> **Similarity to GAIL:** while GAIL and our algorithm for approximating the BPD both learn generative models of human behavior, they are fundamentally different—GAIL and the BPD solve fundamentally different problems. GAIL aims to imitate behavior (i.e. it takes demonstrated trajectories and produces a single policy), while the BPD generates a distribution over possible behavior given a reward function (i.e., it takes a reward function and produces a distribution over policies). While both algorithms leverage a discriminator, the discriminator is also performing very different roles in each. In GAIL, the discriminator aims to bring the imitation policy closer to the demonstration trajectories, while in the BPD, the discriminator aims to increase the entropy of the distribution over policies to ensure diversity. This also means that in the two algorithms the discriminators operate on different types of inputs: in GAIL, it takes as input a single state-action pair, while in the BPD it takes as input an entire policy.
>
> GAIL is actually much more similar to behavior cloning (BC), since both are imitation learning algorithms which require extensive human data to train. The reviewer is right in that GAIL makes for a stronger baseline than BC, due to the added robustness from RL. Below, we have thus included results comparing BC and GAIL, which require extra data, and the BPD, which adapts to predict human behavior over a single trajectory without requiring any prior human data. For each method, the cross entropy (lower is better) over test data is shown. We have also included these results in Appendix C.3 of the paper.
>
> Cross entropy on test set (random prediction = 1.79):
>
> |  | Cramped Room | Forced Coordination | Coordination Ring |
> |---|---|---|---|
> | BC | 0.86 ± 0.09 | 1.02 ± 0.07 | 1.15 ± 0.06 |
> | GAIL | 1.02 ± 0.05 | 1.23 ± 0.07 | 1.27 ± 0.04 |
> | BPD | 0.99 ± 0.08 | 1.19 ± 0.11 | 1.40 ± 0.05 |
>
> As can be seen from these results, GAIL performs similarly to BC, and, importantly, the BPD also performs comparably to both despite requiring no human training data outside the evaluation episode (which BC and GAIL both require).
>
> **Unfair comparison between BC and the BPD:** the reviewer remarked that the comparison between behavior cloning (BC) and our proposed method, the BPD, might be unfair since BC does not know the reward function. We agree that BPD has the advantage of leveraging the reward function, but in a sense this is exactly the strength it has over BC, that it is able to leverage the reward function (plus the consistency assumption) to produce a model that’s as good as what BC gets from vast amounts of human data. Arguably, BC and the BPD are both useful in different situations. If one has access to a lot of human data in an environment, it might make sense to train a human model using BC. On the other hand, if one knows the reward function for the environment but does not have human data, the BPD is much more useful. Prior to this work, the main reward-based human model from the literature, Boltzmann rationality, failed to accurately predict human behavior given a reward function, as we demonstrate in Overcooked. In contrast, the BPD now allows prediction of human behavior from a reward function just as accurately as BC does from prior human data.
>
> **Figure 1 caption:** the reviewer argued that in the walking-to-work example, observing the human choosing route A could provide information about future behavior under Boltzmann rationality if the reward function is not assumed to be fixed. We have updated the caption to clarify this.
>
> **Typos:** thank you for pointing these out, we have fixed them!

---

### Official Review · Reviewer_9Tye · 2021-11-02

**Correctness:** 4
**Technical Novelty And Significance:** 3
**Empirical Novelty And Significance:** 3
**Recommendation:** 8
**Confidence:** 4

**Main Review:**

Strengths:
- Very well written.
- Contains many insightful examples and figures.
- Most of the statements are very well supported


**Summary Of The Paper:**

The paper addresses the problem of modeling human behavior in a case where their deviation from the optimal behavior is consistent over time, rather than independent (systematic suboptimality). They claim that systematic suboptimality can be modeled by predicting policies rather than trajectories. To this end, they introduce Boltzmann policy distribution (BPD), as an alternative to Boltzman rationality, as a prior over human policies. BPD enables adaption to human policies over time. While in Boltzmann rationality past behavior cannot be used to predict future behavior (since current action is independent of all the previous actions), the paper claims that since humans are consistent one can use past actions to predict future ones which leads to introducing BPD. They consider that human is sampling over policies rather than trajectories which results in the previous actions and states inducing a posterior over policies. By taking the expectation over the posterior they predict the next action. They further use deep generative models in order to sample from the BPD and minimize the KL divergence between the two distributions (P_BPD and the distribution induced by the generative network) to ensure the predicted distribution is close to the BPD.

Through experiments, they demonstrate cases where Boltzmann rationality is not able to predict the behavior of a suboptimal human while BPD can adapt to the behavior over time and predict the correct next action. They also show that their method is able to predict human behavior as well as the baseline while using fewer data.

**Summary Of The Review:**

The paper addresses and motivates an interesting problem and further solves modifies the previously known Boltzmann rationality, introducing Boltmann Policy Distribution, in order to adapt to the human behaviour over time. The technical details of the approach are well written and supported and experiments are well designed to give illustrations of the cases where the Boltmann rationality will fail while BPD manages to predict human behaviour. I believe the paper is a good paper and authors have taken time to write it in an elegant way and design insightful examples to motivate their problem and show the advantages of their method.

---

> ### Author Response · Authors · 2021-11-16
> **Response to Reviewer 9Tye**
>
> We thank the reviewer for their positive comments on our paper. We appreciate that the reviewer thought that our paper is well-written with helpful examples and that our experiments support the effectiveness of the BPD compared to Boltzmann rationality.

---

### Official Review · Reviewer_Cv9m · 2021-11-03

**Correctness:** 4
**Technical Novelty And Significance:** 3
**Empirical Novelty And Significance:** 2
**Recommendation:** 8
**Confidence:** 2

**Main Review:**

The idea is pretty interesting and the approximation makes it possible to train a rather implausible model BPD. The paper is well written. I find it easy and clear to read through, without the need to go back and forth. The hyperparameters and the setup of the experiments are clearly provided. It would be good if the code can be shared if accepted.

Though the model is only tested on one complicated task and more experiments are always better, I find the experiments in the paper serve their purpose to illustrate the effectiveness of the model. I also find the simulation very useful for understanding the model, and as a sanity check to make sure after approximation the model is doing something as expected.

The intuition of the model is sort of based on “people are consistent over time”. It makes sense to me, but I’m wondering if there are any references on this. The better performance of BPD compared to BR sort of supports this, but it’s not clear whether it is the consistency of choice that actually benefits BPD. BPD is clearly a much larger and complicated model.

It would be good to put the size of the training set for BPD compared to BC. Is the BPD only directly trained/adapted on the testing data? It would be good to briefly discuss how easy/hard to train BPD.

**Summary Of The Paper:**

The paper proposes a new model BPD to account for the suboptimality of human behavior, and provide an approximation inference method for BPD. The authors illustrate BPD through a simple simulation and compare BPD with existing methods on the overcooked game. They show that their model can match the performance of the data-extensive BC method and outperform BR. They further evaluate their model on human-AI collaboration and show that their model can match or even outperform the human-aware RL model.

**Summary Of The Review:**

I’m overall positive about this paper. The idea is interesting and the experiments illustrate the effectiveness of the new model. The model has the potentials to be applied and further adapted to other settings.

---

> ### Author Response · Authors · 2021-11-16
> **Response to Reviewer Cv9m**
>
> We thank the reviewer for their positive comments. We appreciate that the reviewer felt our paper was clear and that the experiments illustrated the effectiveness of the BPD. As the reviewer suggested, we will share our code and models if the paper is accepted.
>
> The reviewer suggested that the claim that “people are consistent over time” could use more evidence. This is a well known phenomenon from the behavioral psychology and economics literature [1, 2]; we have added these references to the introduction of the paper to support this claim. As additional evidence, we show in Figure 8 a few examples of how humans playing Overcooked consistently take the same suboptimal actions at different points during an episode.
>
> The reviewer also asked for clarity on the training of the BC policy and the BPD. The BC policy was trained on a training set of human trajectories, while the BPD had no access to this set. Instead, the BPD prior was calculated entirely offline using only the environment and reward function, as shown in Figure 2. Then, for each test episode of human behavior, it was updated with a Bayes filter over the course of that single episode, using only the past actions from that episode to predict each following action (also shown in Figure 2).
>
> Finally, the reviewer mentioned it would be helpful to discuss how easy it is to train the BPD. We generally found that the BPD required a larger-than-usual RL batch size compared to training a single policy, since the BPD is essentially an infinite population of policies. However, besides this, we were able to use the hyperparameters, network architectures, and reward shaping largely unchanged from the original Overcooked paper. Thus, while the BPD is somewhat more computationally demanding than typical RL methods, we did not find it needed much additional hyperparameter tuning. We have included these experiences in Appendix A of the paper.
>
> [1] David C. Funder and C. Randall Colvin. Explorations in Behavioral Consistency: Properties of Persons, Situations, and Behaviors. Journal of Personality and Social Psychology 60, no. 5 (1991): 773–94. https://doi.org/10.1037/0022-3514.60.5.773.
>
> [2] Ryne A. Sherman, Christopher S. Nave, and David C. Funder. “Situational Similarity and Personality Predict Behavioral Consistency.” Journal of Personality and Social Psychology 99, no. 2 (2010): 330–43. https://doi.org/10.1037/a0019796.

---

### Official Review · Reviewer_Hd8J · 2021-11-03

**Correctness:** 4
**Technical Novelty And Significance:** 4
**Empirical Novelty And Significance:** 4
**Recommendation:** 8
**Confidence:** 4

**Main Review:**

Overall, I recommend accepting this paper because it provides a theoretically clean analysis of a novel insight that is relevant to important problems. The key insight is that one can use the Boltzmann distribution over policies as a prior, to make behavioral cloning more data efficient.

Significance:

The proposed approach is computationally expensive, but it makes sense to use it when compute is abundant but human demonstrations are scarce or expensive. This is likely to be a common situation in the future as human time stays equally expensive while computation budgets grow. Hopefully, paper’s insights can spur more computationally efficient approaches.

The paper provides insights relevant mainly to two important issues. Firstly, imitation learning from systematically suboptimal human behavior. Secondly, human-AI coordination, which benefits from predicting human behavior and is increasingly relevant in areas such as robotics and autonomous driving.


Experiments:

The authors empirically demonstrate clear advantages over Boltzmann rationality. They also show that their method better predicts human behavior, and that it is more data-efficient than behavioral cloning.

Aside from a toy environment, the evaluation is done in Overcooked, a complex and interesting environment. Overcooked is fittingly a key benchmark for human-AI collaboration which is a goal of this paper.

The paper is overall very well written and illustrated.


Concerns:

1) The authors left it unclear why BPC should be more data efficient than behavioral cloning (BC). While BPC needs less samples empirically, this could just be because the BPC was allowed to learn online during each episode whereas BC was not. In fact, the BC baseline is left underspecified. Is the proposed method a special case of BC that uses a specific prior (the BPC)? For a fair comparison, the authors should use the same models, inference algorithms, online learning, etc for BC and the proposed method. Arguably, the only difference should be the prior (BPC) or any other aspect that the authors claim is novel.

2) One of the motivations of the paper is to understand human preferences, but the proposed method assumes that the human reward function is already known. This seems contradictory or at best requires explanation. I’m referring to the first sentence in the paper here. If the motivation is not to understand human preferences, I’d suggest to clarify the actual motivations throughout the introduction. The other motivations (predicting human actions and collaborating with humans) make more sense to me so you could simply remove ‘understanding human preferences’ to partially address my comment. However, I also think these motivations deserve more than one sentence of elaboration. Additionally, the conclusion says that the the BPD could be useful for reward learning, even though it assumes that the reward is known. There’s also explanation needed here.

3) A further limitation that should be discussed is that the BPC assumes the human policy to be close to optimal with high probability. If the human is systematically irrational they may achieve a much lower reward than what is optimal. Consequently, the BPC may assign very low probability to the true human policy.





———————— Detailed comments ————————


The experiments don’t seem to answer if the BPD prior is actually needed for good performance. It might be enough to do online inference with a different prior (e.g. uniform)?

“Understanding human preferences, predicting human actions, and collaborating with humans all require models of human behavior. “; “However, Boltzmann rationality fails to capture human behavior that is systematically suboptimal.”: some of these key claims currently lack support in the introduction. They’ve been supported e.g. in arXiv:1712.05812 and arXiv:1512.05832.

Figure 2 lacks a bracket after R(s,a). It also lacks the labels b) and c).

The introduction was a bit long, e.g. the last paragraph does not seem essential.

The paper repeatedly says that BPD is hard to compute because policies lie in a (high-dimensional) continuous space. It’s unclear why continuity would be a problem here, and I suspect this is not true. Indeed a high-dimensional discrete space seems more difficult for optimization and inference.


**Summary Of The Paper:**

The goal of this paper is to train models that imitate human behavior using limited samples. A human reward function is given but human behavior may be suboptimal according to this reward function and therefore we cannot use the standard assumption, Boltzmann rationality. The authors propose an imitation learning algorithm that uses the known reward function to form a prior over human policies, the Boltzmann policy distribution (BPD). A posterior over policies is then inferred from human behavior. This approach is empirically more data efficient than behaviorally cloning human behavior.


**Summary Of The Review:**

Overall, I recommend accepting this paper because it provides a theoretically clean analysis of a novel insight that is relevant to important problems. The key insight is that one can use the Boltzmann distribution over policies as a prior, to make behavioral cloning more data efficient. The experiments and presentation are overall convincing. I have concerns about the behavioral cloning baseline and the relevance to learning preferences.


UPDATE: Although we were not able to resolve some confusion during the discussion period, my substantive comments have been addressed so I have updated my score now.

---

> ### Author Response · Authors · 2021-11-16
> **Response to Reviewer Hd8J**
>
> We thank the reviewer for the extensive comments. We are glad that the reviewer appreciated that our method is theoretically clean and agreed with our experimental analysis suggesting that the BPD is superior to Boltzmann rationality. Below, we have responded to criticisms and questions.
>
> **BC baseline comparison:** we agree that an additional natural baseline is training BC online during a single episode starting from a randomly initialized network. We conducted experiments with this baseline; the cross entropy (lower is better) on the test set for each method is listed in the table below. Note that for all BC experiments in the paper, we used a set of hand-designed features from the original Overcooked paper. In contrast, all experiments with the BPD used the raw observations from the environment. Thus, we actually stacked the cards against the BPD to a certain extent. We have included these results in Appendix C.3.
>
> Cross entropy on test set (random prediction = 1.79):
>
> |  | Cramped Room | Forced Coordination | Coordination Ring |
> |---|---|---|---|
> | Online BC (hand-designed features) | 1.11 ± 0.13 | 1.46 ± 0.17 | 1.94 ± 0.11 |
> | Online BC (raw observations) | 1.42 ± 0.13 | 2.10 ± 0.35 | 2.67 ± 0.22 |
> | BPD (raw observations) | 0.99 ± 0.08 | 1.19 ± 0.11 | 1.40 ± 0.05 |
>
> The BPD is consistently better at predicting real human behavior than running BC online, with an even larger difference when the two algorithms are compared using the same inputs. Besides the BPD's higher accuracy, it might also be preferable because it could be intractable to update a BC model online after every human action, particularly in cooperative settings, where using RL to update the agent’s cooperative policy based on an updated BC human model is very slow. In contrast, as we show, the BPD can be computed entirely offline and used to train cooperative policies which run in real time.
>
> **Connection to preference learning:** we agree that the connection between the BPD and preference learning was not well explained in the paper. Our argument for why the BPD can be useful for preference learning is that any reward-conditioned human model can be inverted to infer a reward function from behavior—and if the BPD is a better model, we’d in turn expect reward inference to be more accurate. That is, suppose one has a human model $p(\tau \mid R)$ that gives a distribution over trajectories $\tau$ given a reward function $R$. Then one can either compute a maximum-likelihood reward function for a given trajectory, i.e. $R^* = \max_R p(\tau \mid R)$, or a Bayesian posterior over reward functions, i.e. $p(R \mid \tau) \propto p(\tau \mid R) p(R)$. These problems are solved by inverse reinforcement learning (IRL) algorithms. Since we show that the BPD better predicts real human behavior than Boltzmann rationality when the reward function is known, we argue that it could also be a more useful human model for IRL. However, this is simply a discussion point about the broader implications of our work, not a specific result we claim. We have updated the last paragraph of the introduction to clarify this.
>
> **Limitation in assuming the human is close to optimal:** this is definitely a limitation of the BPD which is also shared with Boltzmann rationality, and we have clarified it more explicitly in the paper. However, the question of whether people are actually close enough to optimal for the BPD to be useful is an empirical one, and at least in Overcooked we have demonstrated that the BPD is a useful prior for predicting human behavior.
>
> **BPD prior versus uniform prior:** the reviewer mentioned that “it might be enough to do online inference with a different prior (e.g. uniform).” We actually do have results comparing inference with the BPD to inference with a uniform prior in Appendix C.3, using the same inference algorithms for both priors. They suggest that the uniform prior is also useful to some degree for predicting human behavior, but that this is mostly because it accurately predicts that people will “stay” (i.e., take a no-op action) most of the time. When removing such “stay” actions, it fails to perform much better than random prediction.
>
> **Difficulty of inference in continuous spaces:** the reviewer questioned our statement that the continuous (as opposed to discrete) nature of the space of policies makes inference more difficult. We argue that discrete spaces are easier to represent distributions over, because any distribution over a discrete space simply assigns a probability to each discrete possibility in the space. Thus, distributions on discrete spaces are easily parameterized. In contrast, a distribution over a continuous space must be represented by a PDF, which is much more difficult to parameterize in a general manner. Also, it is often easier to sample from a distribution over a discrete space, while it is difficult to flexibly parameterize continuous distributions such that one can both calculate densities and take samples.

---

> > ### Comment · Reviewer_Hd8J · 2021-11-21
> > **Remaining questions**
> >
> > Thank you. I'll possibly update my score after processing the other reviews and replies. I have two remaining questions:
> > 1) Why is it useful that the BPD helps us infer a reward function when the BPD assumes that the reward function is already known? Are you suggesting to transfer your BPD from a known reward function to another, unknown reward function?
> > 2) What is your main argument why BPD is more efficient than BC (especially when given a similar computation budget)? This point seems not very much explained or empirically probed (e.g. with ablations) in the paper. Apologies if I missed something here.

---

> > > ### Author Response · Authors · 2021-11-23
> > > **Answers to remaining questions**
> > >
> > > Thank you for taking the time to read our response and reply. Here are answers to your questions:
> > >
> > > 1. In our paper, the focus is on using a reward function to predict human behavior with the BPD; this remains the primary result of our work. However, we note that in future work the BPD could also be used to solve the inverse problem of inferring a reward function from human behavior, i.e. reward learning. Boltzmann rationality, which we propose replacing with the BPD, is used extensively in reward learning, e.g. in [1-3]. The idea in these papers is to optimize over the reward function which is input to the human model such that the output of the model becomes closer to observed human behavior. In principle, one could do the same thing with the BPD, although we do not propose any concrete algorithms for this. Rather, we leave this as a discussion point about future work.
> > > 2. We do not argue that the BPD is necessarily more computationally efficient than BC; rather, we show that it is more *sample efficient*. In the table above, we showed that training a BC policy over a single minutes-long trajectory of human behavior is much worse at predicting human behavior than using the BPD in the same setting. BC only gives comparable predictive power to the BPD when it has access to far more training data in the form of an offline set of prior human trajectories. Thus, we argue that the BPD allows similar predictive performance with fewer samples, i.e. better *sample complexity*. While the BPD requires more computational power, this is true of many RL-based methods, and, as you mentioned in your original review, “the proposed approach is computationally expensive, but it makes sense to use it when compute is abundant but human demonstrations are scarce or expensive.” We are interested in exactly this situation, which seems realistic given the increasing computational power available to ML. We will clarify this in the paper.
> > >
> > > [1] Ziebart, B. D., Maas, A. L., Bagnell, J. A., & Dey, A. K. (2008). Maximum Entropy Inverse Reinforcement Learning.
> > >
> > > [2] Christiano, P., Leike, J., Brown, T. B., Martic, M., Legg, S., & Amodei, D. (2017). Deep Reinforcement Learning from Human Preferences.
> > >
> > > [3] Hadfield-Menell, D., Milli, S., Abbeel, P., Russell, S., & Dragan, A. (2020). Inverse Reward Design.

---

> > > > ### Comment · Reviewer_Hd8J · 2021-11-28
> > > > **Already known functions**
> > > >
> > > > Thank you.
> > > >
> > > > Re 1: My question was why it is interesting to use the BPD for inverse RL in a setting where the reward function is already known. (As the BPD assumes the reward function is known). Why infer a reward function that is already known?
> > > >
> > > > Re 2: I meant that BC could be might be more data-efficient as well by using more computation than in the particular implementation you chose. E.g. you could update the weights using MCMC instead of gradient descent, and use a model with stronger inductive bias such as a Gaussian process.

---

> > > > > ### Author Response · Authors · 2021-11-30
> > > > > **Additional response**
> > > > >
> > > > > Thanks for your further clarifying questions.
> > > > >
> > > > >  1. You are absolutely right; there is not much use in inferring the reward function via inverse RL if the reward is already known. The BPD (like Boltzmann rationality) is indeed conditioned on a known reward. However, there will be tasks where the reward is not known. There, we discuss that we can do inverse RL using the BPD as a model of human behavior. Concretely, inverse RL uses a model of reward → behavior, and inverts that to infer reward from behavior. Common models are reward → optimal behavior, or reward → Boltzmann distribution. Since the BPD seems to more accurately capture human behavior (per our results), one might get better inference of the reward by inverting the reward -> BPD model instead. We do not provide any concrete algorithms to invert the reward → BPD model but leave it as a discussion point about possible future work.
> > > > >
> > > > >  2. We agree that the online BC baseline might be a weak one compared to more sophisticated Bayesian techniques. We think that a stronger baseline is to assume a uniform prior over policies and then perform Bayesian inference using the same inference algorithms and computational power as used with the BPD. We have results for this baseline in Appendix C.3 and Figure 12. The results show that this uniform prior baseline is indeed better at predicting behavior than basic online BC, but still not as good as the BPD, particularly when only considering predictions over non-noop actions.

---

### Decision · Program_Chairs · 2022-01-20

**Decision:**

Accept (Poster)

**Comment:**

The paper presents a new approach to learning human behavior by observing a small number of interactions. To this end, it proposes a Bayesian learning framework where a Boltzmann-type prior over human policies, based on an available reward function, governs default behavior. The prior is updated by observing actual trajectories taken by (human) agents, in principle. In practice, a full-fledged implementation using Gaussian priors and features from a neural architecture is proposed and shown to be effective in practical benchmarks.

The reviewers are all positive about the paper's contributions. One remaining concern is that the effect of the quality of the prior (Boltzmann vs. other type vs. features designed in a different way) on the learning process is not explored to a significant depth. Yet, the results and approach proposed in the paper are valuable to merit acceptance.